# Cell-TRACTR: A transformer-based model for end-to-end segmentation and tracking of cells

**Owen M. O'Connor**[1,2], **Mary J. Dunlop**[1,2]*

**1** Biomedical Engineering, Boston University, Boston, Massachusetts, United States of America,
**2** Biological Design Center, Boston University, Boston, Massachusetts, United States of America

\* mjdunlop@bu.edu

## Abstract

Deep learning-based methods for identifying and tracking cells within microscopy images have revolutionized the speed and throughput of data analysis. These methods for analyzing biological and medical data have capitalized on advances from the broader computer vision field. However, cell tracking can present unique challenges, with frequent cell division events and the need to track many objects with similar visual appearances complicating analysis. Existing architectures developed for cell tracking based on convolutional neural networks (CNNs) have tended to fall short in managing the spatial and global contextual dependencies that are crucial for tracking cells. To overcome these limitations, we introduce Cell-TRACTR (Transformer with Attention for Cell Tracking and Recognition), a novel deep learning model that uses a transformer-based architecture. Cell-TRACTR operates in an end-to-end manner, simultaneously segmenting and tracking cells without the need for post-processing. Alongside this model, we introduce the Cell-HOTA metric, an extension of the Higher Order Tracking Accuracy (HOTA) metric that we adapted to assess cell division. Cell-HOTA differs from standard cell tracking metrics by offering a balanced and easily interpretable assessment of detection, association, and division accuracy. We test our Cell-TRACTR model on datasets of bacteria growing within a defined microfluidic geometry and mammalian cells growing freely in two dimensions. Our results demonstrate that Cell-TRACTR exhibits strong performance in tracking and division accuracy compared to state-of-the-art algorithms, while also meeting traditional benchmarks in detection accuracy. This work establishes a new framework for employing transformer-based models in cell segmentation and tracking.

### Author summary

Understanding the growth, movement, and gene expression dynamics of individual cells is critical for studies in a wide range of areas, from antibiotic resistance to cancer. Monitoring individual cells can reveal unique insights that are

**Data availability statement:** Code Availability We provide the code for Cell-TRACTR at https://gitlab.com/dunloplab/Cell-TRACTR. Code for Cell-HOTA is available at https://gitlab.com/dunloplab/Cell-HOTA. Model checkpoints are available on Zenodo (https://zenodo.org/records/14509424). Datasets The bacterial mother machine dataset is hosted on Zenodo (https://doi.org/10.5281/zenodo.11237127). It is formatted in the style of the Cell Tracking Challenge. The DeepCell dataset was downloaded directly from the DeepCell website (https://datasets.deepcell.org/data). We used their "DynamicNuclearNet Tracking" dataset. This dataset was converted into the Cell Tracking Challenge format to be compatible for training the models.

**Funding:** This work was supported by the National Science Foundation (MCB-2143289 and MCB-2032357 to MJD). The funders had no role in study design, data collection and analysis, decision to publish, or preparation of the manuscript.

**Competing interests:** The authors have declared that no competing interests exist.

obscured by population averages. Although modern microscopy techniques have vastly improved researchers' ability to collect data, tracking individual cells over time remains a challenge, particularly due to complexities such as cell division and non-linear cell movements. To address this, we developed a new transformer-based model called Cell-TRACTR that can segment and track single cells without the need for post-processing. The strength of the transformer architecture lies in its attention mechanism, which integrates global context. Attention makes this model particularly well suited for tracking cells across a sequence of images. In addition to the Cell-TRACTR model, we introduce a new metric, Cell-HOTA, to evaluate tracking algorithms in terms of detection, association, and division accuracies. The metric breaks down performance into sub-metrics, helping researchers pinpoint the strengths and weaknesses of their tracking algorithms. When compared to state-of-the-art algorithms, Cell-TRACTR meets or exceeds many current benchmarks, offering excellent potential as a new tool for the analysis of series of images with single-cell resolution.

## Introduction

Single-cell resolution experiments have significantly enhanced our understanding of cell physiology and survival in the context of a wide range of domains including antibiotic resistance [1], cancer [2], stress response [3,4], and aging [5,6]. Traditional population-level measurements can obscure heterogeneity among cells such as single-cell differences in gene expression that can emerge across space or time. Because this variation can be stochastic in origin [7], it is often necessary for researchers to gather large datasets to achieve statistically significant insights. Advances in automated imaging, coupled with increasing computer storage capacity, have dramatically increased the throughput of data collection. However, despite the abundance of data, analysis can remain a bottleneck, particularly for applications with many sequential images acquired over time. Although cell segmentation algorithms have achieved impressive performance levels [8–11], cell tracking still presents a significant challenge.

Cell tracking datasets are often markedly different from standard multi-object tracking datasets. In contrast to classic computer vision datasets like those that feature human subjects or automobiles moving through a field of view, cells can divide, and division events can occur frequently during the time frame of an experiment. Moreover, cells often display a high degree of similarity in appearance, are small in resolution, and can morph in shape over time. In addition, time-lapse microscopy experiments, which can extend for hours or even days, may have a low frame rate to reduce the impact of imaging on cells. However, the reduced frame rate can complicate tracking due to the non-linear motion often exhibited by cells. For example, growth of a microcolony may produce unpredictable motion of an individual cell, in contrast to images involving pedestrians or vehicle motion, which tend to follow more predictable patterns. In addition, shifts in the field of view can occur due to

changes in imaging conditions over time. Further complicating analysis, microscopy images can contain a high density of cells, especially for applications that involve the analysis of cellular communication or community dynamics [12,13]. Thus, cell tracking algorithms must contend with the complexities of cellular behavior coupled with the technical constraints of microscopy and live cell imaging.

While early cell tracking models employed non-machine learning techniques [14–17], the landscape has shifted dramatically towards the adoption of deep learning-based methods [18]. The acceleration of AI in the past decade has ushered in a new era within the field of cell tracking, with deep learning models surpassing simpler machine learning approaches such as nearest-neighbor algorithms [19]. Traditionally, most deep learning algorithms have employed a tracking-by-detection approach where segmentation and tracking are performed in two separate steps [14,20–22]. However, a recent trend in the field is a move towards models where segmentation and tracking are performed in one step [23–29]. Combining segmentation and tracking into one step can help ensure coherent predictions, where tracking information can enhance segmentation accuracy. For example, temporal information helps the segmentation model understand that cells can divide but not merge. Furthermore, some models now incorporate long-term temporal information, providing more context for segmentation [23,28]. While these improvements have demonstrated superior performance compared to models that utilize the tracking-by-detection approach, these models typically still require separate decoders for segmentation and tracking [23,24] or rely on post-processing [27,28]. A unified decoder could enhance the synchronization of segmentation and tracking, leading to more consistent predictions, while eliminating post-processing steps would simplify workflows and improve model generalizability.

In cell tracking and segmentation software, convolutional neural networks (CNN) are currently the preferred method, evidenced by their widespread adoption within the field [10,23–25,27,28]. In particular, U-Net [30] and its derivatives have been used extensively for cell tracking [10,20,25,26,28] and cell segmentation [8,10,31,32], demonstrating the versatility of this architecture. However, deep learning approaches in cell tracking also extend beyond CNNs. Other architectures, such as graph neural networks [33,34], reinforcement learning [35], recurrent neural networks [22,25], and unsupervised learning [19] approaches, have also been successfully employed. Despite these advances, the utilization of transformers [36], which have revolutionized many areas of machine learning including computer vision [37,38], remains relatively underexplored in cell tracking. Transformers use an attention mechanism that can effectively model the relationships between different parts of an image, making it easier to follow the same object across frames. This attention mechanism has demonstrated great success in cell segmentation, where transformers have made significant strides [9,11,39,40]. While emerging work, like Trackastra [41], has demonstrated the promising potential of transformers in cell tracking, the significant capabilities of transformers, as demonstrated by the state-of-the-art multi-object tracking algorithm MOTRv3 [42], have not been fully leveraged for cell tracking. We were inspired by the top performance of MOTRv3 on the DanceTrack dataset [43], which features objects (dancers) with similar appearances and diverse motion patterns. This tracking dataset closely parallels cell tracking, where cells can have nearly identical appearances and migrate in non-linear directions. We reasoned that top performing algorithms on the DanceTrack dataset, like MOTRv3, would be well-suited to track cells.

To leverage these capabilities, here we introduce a novel approach to cell segmentation and tracking that uses a transformer-based model to enhance tracking performance. Inspired by TrackFormer [44] and MOTR [45], which are multi-object tracking algorithms from the broader computer vision community, we developed a transformer-based model that can simultaneously segment and track dividing cells. This model builds upon the framework of the Detection Transformer (DETR) [46] which was the first deep learning model to perform end-to-end object detection without the need for any post-processing like non-maximum suppression. Although DETR demonstrates remarkable performance, it requires longer training periods and greater computational resources than CNN-based approaches.

The evolution of DETR has led to significant enhancements in terms of accuracy and reduced training times through innovations such as deformable attention [47], dynamic anchor boxes [48], contrastive denoised training [49,50], and

query selection [51,52]. Among these innovations, Mask-DINO [53] and Co-DETR [54] have emerged as state-of-the-art models for instance segmentation and object detection, respectively. Meanwhile, TrackFormer [44] and MOTR [45] have been instrumental in advancing tracking capabilities for DETR-based models. Recent iterations have built upon this foundation, offering further improvements [42,55–59], culminating in the latest iteration of MOTR, called MOTRv3 [42], which has established itself as a state-of-the-art model in multi-object tracking.

Historically, cell tracking algorithms used datasets from the Cell Tracking Challenge [60] to benchmark performance. However, a notable portion of researchers have opted to use custom datasets or the increasingly popular DeepCell dataset [22]. Although the Cell Tracking Challenge has provided a useful way to standardize evaluations, it has drawbacks since most of the datasets are limited in the number of images and are only partially annotated, posing challenges for deep learning models that can require large, fully annotated datasets. The DeepCell dataset addresses this by providing a large and diverse dataset complete with full annotations.

In addition to providing datasets, the Cell Tracking Challenge also developed associated metrics to quantify cell segmentation and tracking accuracy. These metrics have been widely adopted for assessing model performance, however recent critiques, especially regarding their handling of cell division, have prompted researchers to modify these metrics or add new metrics [22,23,61]. Similarly, we found that the tracking metric from the Cell Tracking Challenge tends to de-emphasize division accuracy while over-emphasizing segmentation precision. Meanwhile, the general multi-object tracking community has adopted the Higher Order Tracking Accuracy (HOTA) metric due to its balanced assessment between tracking and segmentation. HOTA offers interpretable results that can help identify where a model is under performing, however the classic version of this metric does not handle division events.

In this work, we introduce Cell-TRACTR, a new transformer-based model to segment and track cells. The model can simultaneously segment and track without requiring any post-processing and it can handle frequent division events. Transformers offer a powerful architecture to handle the difficult challenge of tracking cells. In addition, we introduce a new cell tracking metric which adapts the HOTA metric [62] to effectively account for cell division. Cell-HOTA assesses overall tracking performance by balancing the algorithm's performance on detection, association, and localization. This metric can be broken down into interpretable sub-metrics, providing useful insight into sources of error. We compare our new transformer-based model (Cell-TRACTR) to other state-of-the-art models in the cell tracking field (DeLTA [10,20], EmbedTrack [24], Trackastra [41], and Caliban [22]). We evaluate these models on both bacterial and mammalian datasets. Our results demonstrate Cell-TRACTR's excellent performance in cell segmentation and tracking, achieving accuracy on par with state-of-the-art cell tracking algorithms. Additionally, we illustrate the interpretability of the Cell-HOTA metric for evaluating tracking outcomes.

## Results

### Cell-TRACTR simultaneously segments and tracks cells

Our overall goal was to achieve accurate segmentation and tracking of cells given time-lapse microscopy data. We tested our algorithm on two types of data: microscopy images of *Escherichia coli* growing in the "mother machine," [63] a widely used microfluidic device where bacteria are constrained to grow in single-file lines (Fig 1A), and mammalian cells with stained nuclei growing in a two-dimensional framework (Fig 1B) [22]. These types of experiments can generate large amounts of data, motivating the need for automated image analysis. For example, the mother machine device contains thousands of parallel chambers, each of which is imaged over many hours. Cells growing in two dimensions present additional challenges, where they may move in any direction and morph in shape over time.

We developed Cell-TRACTR by employing a DETR-based framework that couples a CNN backbone with a transformer encoder-decoder architecture [46] (Fig 1C). This end-to-end model, where the input is the raw image and the outputs are embeddings that represent the class and position of each cell in the frame, eliminates the need for post-processing techniques like non-maximum suppression that are required for other tracking-by-detection approaches [64–66]. When an image is input into the model, we first use a CNN to preprocess the image into a more compact and rich representation

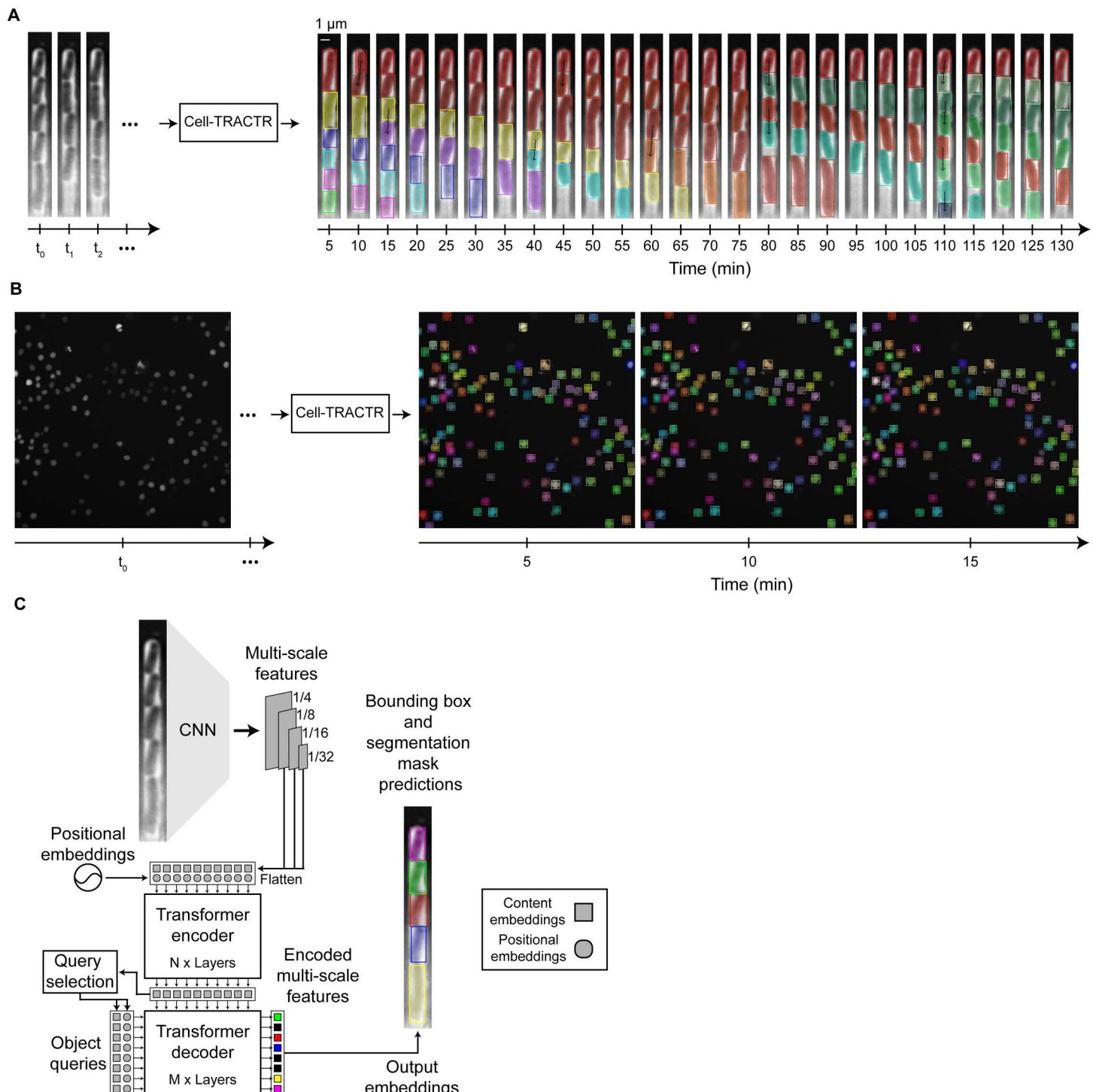

**Fig 1. Results from Cell-TRACTR and model overview for object detection.** (A) Representative output from Cell-TRACTR for a single chamber from a time-lapse microscopy movie of *E. coli* bacteria growing in the mother machine microfluidic device. The output is displayed as a kymograph where the same chamber is shown over time. Each color represents a unique cell being tracked and black arrows signify cell divisions. This movie is from the test set. (B) Overview of Cell-TRACTR tracking mammalian nuclei across multiple frames. Each color represents a unique cell being tracked. These frames are from the DeepCell dataset using data from the test set. (C) Cell-TRACTR is a DETR-based model that first uses a CNN backbone to extract image features, then uses a transformer encoder-decoder framework that leverages attention mechanisms. Encoded multi-scale features are used for query

selection to initialize the object queries. Object queries attend to the encoded multi-scale features to generate the output embeddings needed to predict class labels, bounding boxes, and segmentation masks. If the model predicts an object query to be a cell, then the output embedding is shown as a colored box in the schematic. Object queries predicted to be of class "no object" are colored black and are discarded. The squares represent content embeddings and the circles represent positional embeddings.

since it would be computationally prohibitive to process the full image directly with a transformer. The goal of the CNN is to produce multi-scale features that can be fed into the transformer. Multi-scale features consist of the final layers that are output from the CNN, and they capture properties such as edges, colors, and simple textures. We adopted a ResNet [67] CNN architecture as the backbone to minimize model size while preserving high performance.

## Cell-TRACTR architecture: Cell segmentation

After the CNN extracts information from the input image, an encoder further refines these features, preparing them for the decoder. The multi-scale features along with their positional embeddings are fed into the encoder. The multi-scale features serve as content embeddings which are latent vector spaces that store semantic information about the image. The positional embeddings are latent vector spaces that supply spatial context to the multi-scale features, compensating for the transformer's inherent lack of spatial processing capability. In the encoder, deformable self-attention is used to further enhance the multi-scale features, following the approach in Deformable DETR [47]. The model next performs "query selection," also called "two stage" in [47], which aligns with the Deformable DETR framework, to convert the encoded multi-scale features into region proposals that are used to initialize the object queries (S1 Fig and S1 Text). Briefly, for each encoded multi-scale feature, a class label, bounding box, and segmentation mask prediction are generated. The class label represents the category of an object ('cell' or 'no object') while the bounding box represents the normalized center coordinates and the height and width of the box with respect to the input image. The segmentation mask is an image which represents the pixels associated with each predicted cell. These masks are generated by integrating information from the output embeddings, the encoded multi-scale features, and the original multi-scale features; this process of mask generation is described in further detail below after tracking and cell division are introduced. The top-K encoded multi-scale features are selected based on the highest values of the class labels (S1 Fig). Bounding boxes, extracted from the top-K segmentation masks, are used as the positional embeddings and the top-K encoded multi-scale features are used as the content embeddings. K is determined by the number of the object queries. Combining these positional and content embeddings forms the object queries which are passed to the decoder.

Within the decoder, the object queries interact with the encoded multi-scale features to generate output embeddings that are used to refine an object's bounding box and predict the segmentation mask. The object queries first perform self-attention to avoid duplicate predictions and then deformable cross-attention, where the object queries spatially attend to the encoded multi-scale features (S1 Fig). Output embeddings that are not predicted to be cells (i.e., their class label is 'no object') are discarded. Thus, the final output of the decoder is a set of output embeddings, where each embedding encodes information about the class label, and location of each predicted cell within the image given by both a bounding box that defines its location and a segmentation mask that defines the pixels associated with the cell.

## Cell-TRACTR architecture: Cell tracking

We next leveraged the output embeddings to track cells from frame to frame using the same network architecture. We developed a model that tracks cells over multiple frames using a transformer architecture (Fig 2A). We based the tracking scheme on the architecture of TrackFormer [44] and MOTR [45], which are DETR derivatives, where output embeddings are converted into track queries for analysis of the subsequent movie frame (Fig 2B). At time $t_0$, the model performs object detection and locates all cells in the image. For all subsequent frames, the output embeddings associated with each

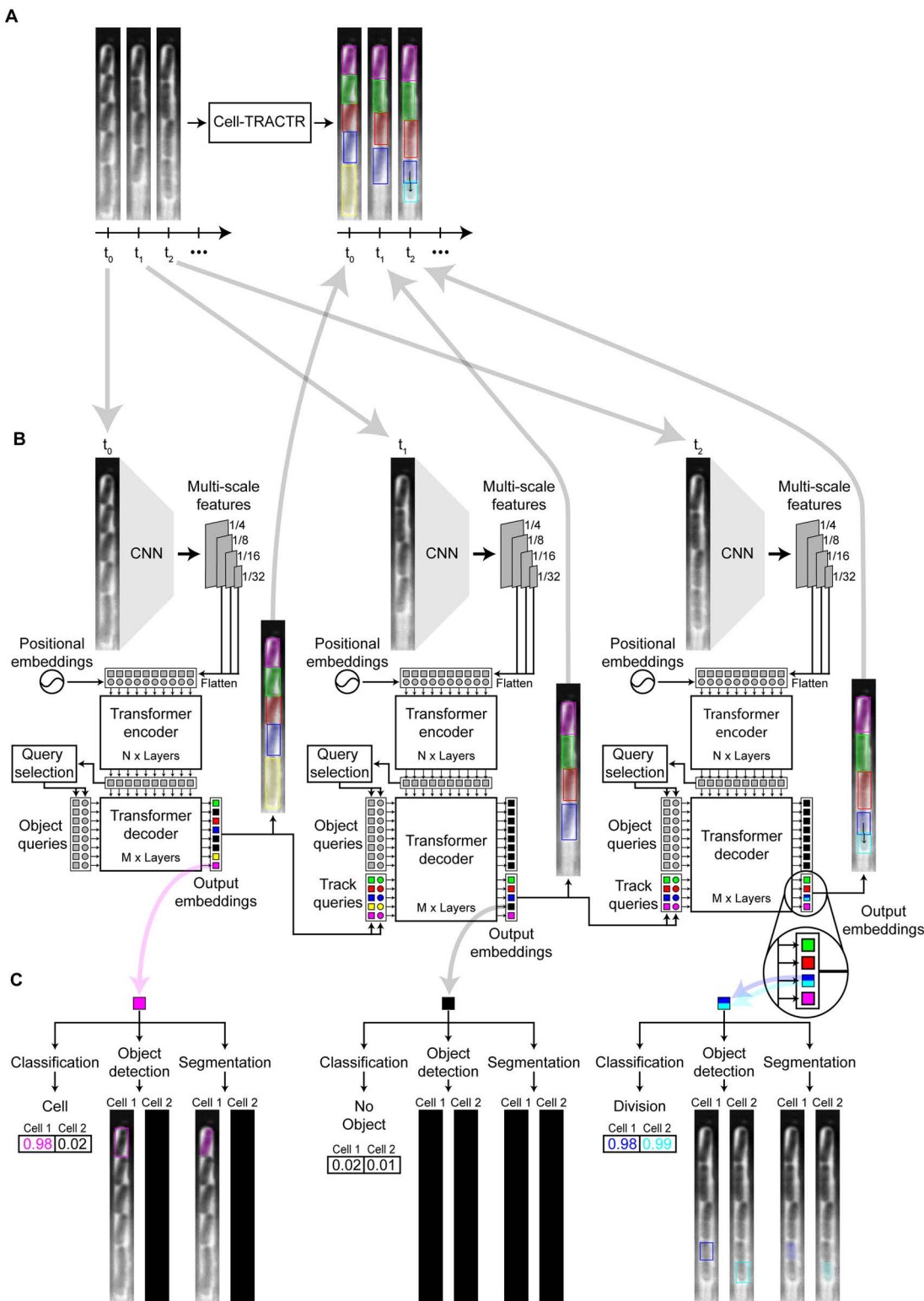

**Fig 2. Visual representation of how Cell-TRACTR uses track queries to track cells and detect divisions.** (A) Overview of Cell-TRACTR performing cell tracking, where a black arrow connecting two cells signifies a cell division event, as seen in frame $t_2$. (B) In frame $t_0$, the model performs object detection and locates all cells in the frame. In subsequent frames, the output embeddings serve as track queries, which track the locations of the cells

from frame to frame. An output embedding split with two colors indicates that the model predicted a cell division. A black output embedding indicates when a cell, represented by a track query, is lost, such as when it exits the field of view. (C) Each output embedding makes two predictions. Each prediction can be classified either as a cell or no object. The class labels determine which prediction is used. If only the first prediction for an output embedding is classified as a cell, then just the first prediction is used. We ignore predictions from output embeddings that predict no object for both class labels. When both predictions are classified as cells, the output embedding is labeled as a division.

detected cell from the previous frame serve as track queries. Track queries predicted as 'no object', which represent cells that exited the field of view, are discarded. Track queries that are predicted to be cells are then concatenated with the object queries of the present frame and fed to the decoder. Self-attention is a critical step in the decoder where track and object queries need to communicate so that an object query does not detect a cell that is already being tracked (S2 Fig). For example, when a cell enters the field the view, an object query is needed to detect the new cell. However, if the cell existed in the previous frame, it needs to be handled by a track query and not an object query. In addition, if a track query makes an error and misses a cell, an object query is needed to detect the missed cell in the next frame. In these scenarios, it is important that the object and track queries can communicate to avoid duplicate predictions. By utilizing object and track queries, Cell-TRACTR simultaneously segments and tracks cells, leading to more accurate and coherent predictions.

### Cell-TRACTR architecture: Cell division

Cell division is an important property that needs to be tracked accurately in single-cell imaging experiments. When a division occurs, as shown in the frame at time $t_2$, the output embedding must record a division event, and this is represented schematically by a split light/dark blue output embedding (Fig 2B). For each output embedding, the model makes a class label, bounding box, and segmentation mask prediction (Fig 2C). For each of these tasks, two predictions are generated. The first prediction corresponds to the cell that is being detected or tracked. The second prediction corresponds to a potential daughter cell generated in a division event, which is only applicable for track queries since object queries can detect at most one cell. The classification branch uses a linear layer to classify each output embedding as 'cell' or 'no object' (S3 Fig). An output embedding is classified as a 'division' when both outputs are classified as 'cell'. The object detection branch uses a multi-layer perception (MLP) to generate a bounding box prediction for each cell. In addition, the segmentation branch produces a segmentation mask for each cell. Closely following the architecture of Mask-DINO [53], multi-scale features are combined with the largest encoded multi-scale feature map to form a pixel embedding map (S3 Fig). The dot product between each output embedding and the pixel embedding map is used to generate the segmentation masks. This process assigns a value ranging from 0 to 1 to every pixel within the mask, indicating the likelihood of each pixel being part of a segmented cell. The maximum value attained by any pixel across all generated masks is combined into one unified mask that encapsulates all predictions. Pixels with values greater than 0.5 in this final mask are classified as belonging to a segmented object. If an output embedding is not predicted to be a cell (i.e., it has a class label of 'no object'), then the associated bounding boxes and segmentation masks are disregarded. Using a transformer-based model allows for seamless segmentation and tracking of cells in a unified step.

As an example, we used Cell-TRACTR to process a time-lapse microscopy movie of *E. coli* growing in the mother machine device (S1 Movie). In the first frame, the object queries detect all cells in the chamber. Then, these object queries are converted into track queries and tracked throughout the rest of the movie. These data provide an example demonstrating that Cell-TRACTR can accurately detect cells, track cells, and predict cell divisions over the course of many generations of growth.

### Multi-object tracking and cell tracking metrics

To assess the performance of Cell-TRACTR, we developed an extended version of the Higher Order Tracking Accuracy (HOTA) metric [62]. Multi-object tracking requires the detection and association of several objects over consecutive frames. When evaluating multi-object tracking, it is important to consider a model's performance across a range of tasks

including the detection, association, and localization of objects. Various tracking metrics have been proposed, but many metrics like Multiple Object Tracking Accuracy (MOTA) [68] and ID Switches (IDS) [69] have been shown to disproportionately emphasize either detection or association [62,70]. An ideal tracking metric should value both detection and association, providing an accurate assessment of the overall performance. Recently, Higher Order Tracking Accuracy (HOTA) [62] has emerged as a leading metric for assessing tracking accuracy due to its balanced approach towards assessing detection, association, and localization. However, most multi-object tracking metrics from computer vision, including HOTA, are designed for objects that do not divide, which is a major limitation for cell tracking applications where division events can be common. In this work, we extended the HOTA metric to include division in its evaluation.

Other approaches exist for measuring the overall performance of cell tracking algorithms. For example, the Cell Tracking Challenge developed a widely used metric, $OP_{CTB}$ (overall performance for the cell tracking benchmark). $OP_{CTB}$ is calculated as the mean of the tracking accuracy measure (TRA) and segmentation accuracy measure (SEG), $OP_{CTB} = 0.5 * (TRA + SEG)$, where tracking is assessed using a graph matching-based approach and segmentation is assessed using the Jaccard Similarity index (S1 Text). Although $OP_{CTB}$ is widely used, recently it has been criticized for not properly handling cell division [22,23,61]. In some cases, removing division links can increase the tracking score. For example, a division event predicted one frame early or late with respect to the ground truth can result in a lower score than removing the division all together (S1 Text and S4 Fig). This is a problem because in real microscopy data it can be challenging to pinpoint the precise frame when a cell divides, and this issue is exacerbated in the case of rapidly growing cells where there are many division events (Fig 3A).

### Cell-HOTA metric

We sought to take advantage of the benefits of the HOTA metric—namely its ability to assess performance in a balanced manner across a range of segmentation and tracking associated tasks—while extending it to handle cell division events. Further, because HOTA is the emerging standard in the broader multi-object tracking community, we wished to align the cell tracking community more closely with this larger group.

The HOTA metric is composed of two main sub-metrics for evaluating tracking: detection accuracy (DetA) and association accuracy (AssA). DetA measures detection accuracy by comparing the number of correct object detections to the total number of detections made. AssA evaluates how closely the predicted paths of objects match their actual paths, averaged over the entire set of correctly detected objects. At a high level, DetA is focused on assessing segmentation accuracy and AssA is specifically tailored to measure tracking accuracy. HOTA incorporates localization accuracy into the score by measuring DetA and AssA across varying similarity thresholds $\alpha$ (Fig 3B). The intersection over union (IOU) between a predicted object and a ground truth object needs to exceed $\alpha$ for the two objects to be considered a match, thus the $\alpha$ value determines the stringency for these matches. HOTA is measured at various values of $\alpha$ to provide insight into how stringency in matching affects DetA and AssA. To improve the readability and follow the terminology used in HOTA, we define a 'tracker cell' as a cell predicted by the tracking algorithm and a 'ground truth cell' as a cell defined in the ground truth.

There are several advantages that HOTA offers compared to $OP_{CTB}$ (S1 Text). First, HOTA is a comprehensive standalone metric that can rank tracking algorithms and can also be decomposed into sub-metrics, allowing for straightforward analysis of performance. Second, HOTA integrates localization accuracy by calculating the average score across all HOTAα scores for each value of α. This is important because it allows the user to assess the detection and tracking performance at the level of stringency that is appropriate for their application. Third, HOTA functions as a monotonic metric with DetA and AssA, meaning an improvement in DetA or AssA will always increase the final HOTA score. Overall, HOTA presents a robust framework for evaluating and ranking tracking algorithms.

To adapt HOTA for cell tracking, we introduced a metric for division accuracy (DivA) alongside the scores for detection and association accuracy. We call this modified metric Cell-HOTA. DivAα is the mean division accuracy score across all cell divisions at a similarity threshold $\alpha$. A true positive division (TPD) is defined by three criteria (Fig 3C): First, two

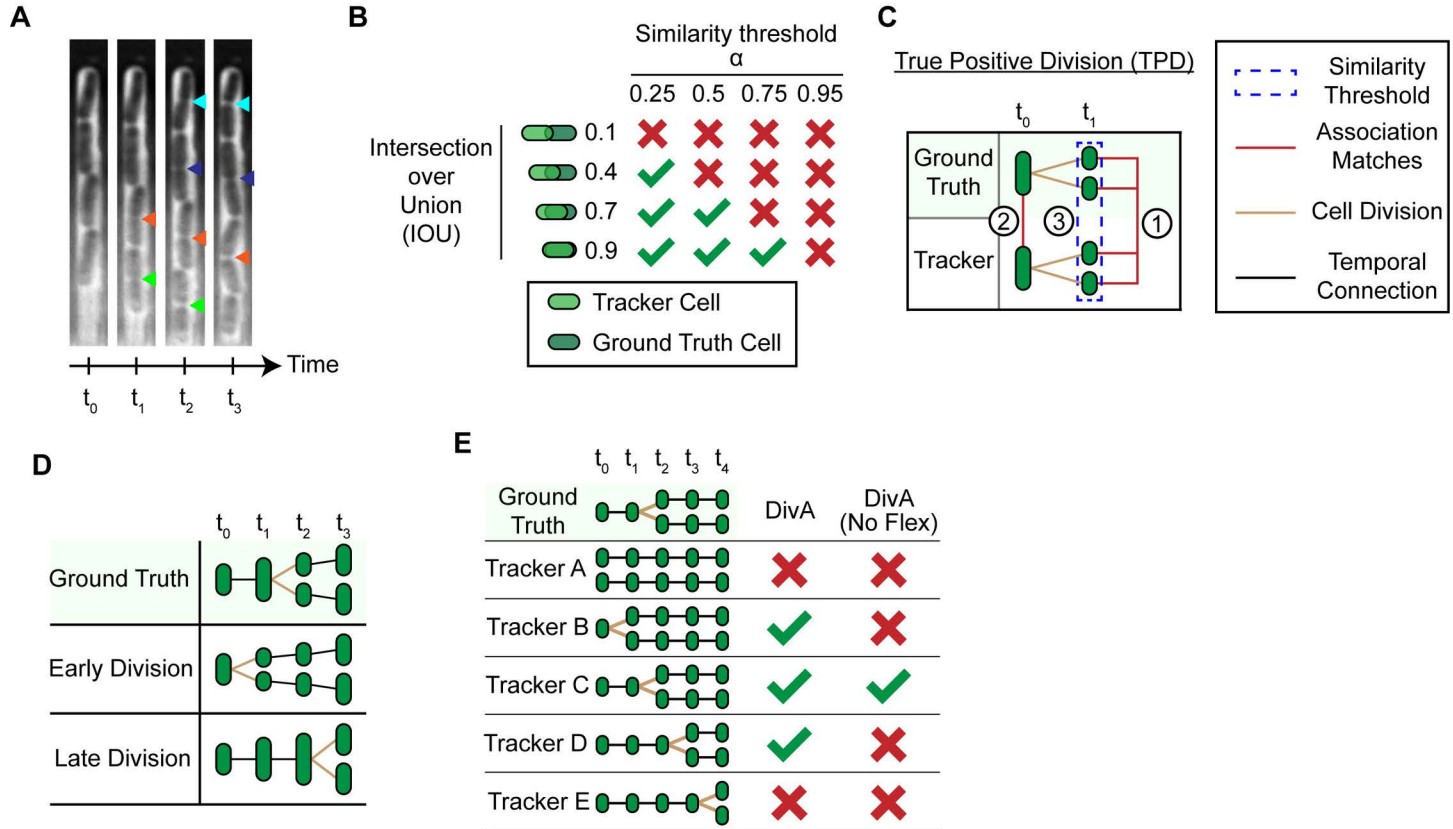

**Fig 3. Cell-HOTA metric incorporates cell division accuracy.** (A) Kymograph of bacteria growing in the mother machine. Colored triangles indicate potential times at which a cell division could have occurred, where different colors correspond to different cells. Multiple triangles of the same color indicate that it can be subjective to determine where division occurs. (B) The intersection over union (IOU) between a tracker cell and ground truth cell must exceed the similarity threshold α to be considered a match. HOTA is evaluated across various values of α, with a higher α imposing stricter criteria for matching. (C) Schematic showing the criteria required to determine if a division qualifies as a true positive division (TPD). A tracker cell and ground truth cell are shown dividing at time $t_1$. First, the set of daughter cells that divided at time $t_1$ needs to match. Second, the parent cells of those daughter cells at time $t_0$ needs to match. Third, the average IOU between the set of daughter cells needs to exceed the similarity threshold α. Red lines indicate matching cells. The blue dotted line indicates the set of cells used to calculate the IOU to determine if the division is a TPD. (D) An early or late division occurs when a cell division is predicted one frame too early or late. Black lines indicate temporal associations and brown lines are used to highlight division events. (E) DivA rewards division events within a frame of the ground truth. Flexible divisions may be disabled so that divisions events are only counted as correct if they occur in the same frame as the ground truth. No Flex indicates that DivA has disabled flexible divisions.

divided tracker cells with the same parent cell need to match with two divided ground truth cells with that same parent cell. Second, the parents of the divided tracker and ground truth cells need to match in the previous frame. Finally, the average IOU between the two matches needs to be greater than *α*. A false positive division (FPD) occurs when a tracker division does not match with a ground truth division. A false negative (FND) division occurs when a ground truth division does not match with a tracker cell division.

Due to the importance of tracking cell divisions in time-lapse microscopy experiments, we set DivAα to be of equal importance to the association score AssAα. AssDivAα is the geometric mean of AssAα and DivAα and we define Cell-HOTAα to be equal to the geometric mean of DetAα and AssDivAα. Similarly to HOTA, Cell-HOTA is a calculated by taking the average Cell-HOTAα score across all values of α. The Cell-HOTA metric is a single unified score that can also be broken down into multiple informative sub-metrics. This makes it easy to analyze whether the algorithm had issues detecting cells, tracking cells, or predicting divisions.

## Allowing for flexibility in determining time of cell division

Another key consideration for cell tracking metrics is that the ground truth is often a subjective assessment of the cell division time rather than an unequivocal record of this event (Fig 3A). For cells that divide frequently, such as in time-lapse images of bacteria, these issues can compound and result in inaccurate estimates of tracking performance. Ultimately, we want to measure a model's capability to generate a coherent cell track rather than replicate the precise ground truths. There are two possible scenarios associated with data where the precise division timing is uncertain and the ground truth is a close, but not exact, record of cell division. We define these alternatives as 'early division' and 'late division' (Fig 3D). In early division, the tracking algorithm predicts that a division event occurs before it does in the ground truth, while in late division it occurs after. In both cases, the overall lineage is preserved outside of this time window. To accommodate the flexible timing of divisions, we modified the DetA$\alpha$, AssA$\alpha$, and DivA$\alpha$ to handle early or late divisions that were one frame away from the ground truth (Fig 3E). In our analysis, we assume any division event within one frame of the ground truth is acceptable, while any division event more than one frame away from the ground truth is erroneous. The flexible division feature within Cell-HOTA is optional and may be turned off. More details on how Cell-HOTA handles flexible divisions are provided in the Methods.

## Benchmarking Cell-TRACTR and other algorithms

To benchmark the performance of Cell-TRACTR, we compared it to three other deep learning-based models and analyzed performance on the same datasets. The first model we benchmarked against was the Deep Learning for Time-Lapse Analysis (DeLTA) [10] algorithm we developed previously. DeLTA consists of two CNNs based on the U-Net [30] architecture, where the two CNNs are used sequentially for segmentation and tracking. DeLTA is a popular model for the analysis of single-cell time-lapse movies of growing bacteria. The second model we benchmarked against was EmbedTrack [24], which is a CNN consisting of a shared encoder, two decoders for segmentation, and one decoder for tracking. We chose EmbedTrack due to its high performance on the Cell Tracking Challenge. The third model we benchmarked against was Trackastra [41], which consists of a transformer decoder specifically designed for cell tracking. Unlike the other models, Trackastra only performs tracking, requiring precomputed segmentation masks to function. We provided Trackastra with the highest-quality segmentation masks (which we determined to be those we derived from our results using EmbedTrack) to maximize its tracking performance. However, it is important to note that the segmentation mask quality directly influences Cell-HOTA and OP$_{CTB}$ scores, which may affect comparison across models.

## Analyzing bacterial mother machine movies

We first trained the four models (Cell-TRACTR, DeLTA, EmbedTrack, Trackastra) on a dataset consisting of 73 time-lapse microscopy movies of bacteria growing in the mother machine. The dataset was split 80%/20% into training and validation data. We trained each model on the training set, tuned using the validation set, and then evaluated on a separate test set which consisted of 29 time-lapse microscopy movies. As a note, we initially tested Trackastra's "General 2D" pre-trained model, but it performed poorly on the test data. Therefore, we trained a model specifically for the bacterial mother machine dataset. Since Trackastra only performs tracking, our evaluation focuses on its tracking performance. Its detection performance directly reflects the quality of segmentation masks provided by EmbedTrack.

When analyzing Cell-HOTA$\alpha$ at various similarity thresholds $\alpha$, we found that all four algorithms performed well on the dataset, especially at lower values of $\alpha$ (Fig 4A). However, there were nuanced differences in performance, with Cell-TRACTR and Trackastra exhibiting superior performance for all $\alpha < 0.75$. When $\alpha > 0.8$, Trackastra, EmbedTrack, and DeLTA's performance slightly exceed Cell-TRACTR, but all dropped off rapidly at high values of $\alpha$. This rapid decrease is expected given the stringency of the similarity threshold in this regime. Overall, we found that all models exhibit similar performance, with Cell-TRACTR surpassing EmbedTrack and DeLTA in overall tracking performance, albeit with slightly less localization precision.

To gain a comprehensive understanding of model performance, we next looked at the sub-metrics of Cell-HOTA. To report a single number, it is common to see the metrics evaluated either at a specific value of $\alpha$ or averaged across all values of $\alpha$. We first selected a single $\alpha$, choosing $\alpha = 0.5$ because it is a typical value used in the literature [53,71]. Cell-TRACTR had the highest overall tracking performance as indicated by Cell-HOTA$_{0.5}$ (Fig 4B). Decomposing this into sub-metrics, we found that all four models have comparable performance for DetA$_{0.5}$, with EmbedTrack slightly outperforming the other models. In contrast, Cell-TRACTR performs well on tracking and cell division, scoring the highest for AssA$_{0.5}$ and DivA$_{0.5}$ at levels that exceed the performance of the other three algorithms. This indicates that while all models exhibit strong detection accuracy, Cell-TRACTR performs very well on tracking and division, while EmbedTrack has the most precise segmentation. More generally, we found that Cell-TRACTR exhibited performance that was on par with current state-of-the-art algorithms, and although there were minor differences between the performance of the individual algorithms, all performed well. These results demonstrate that the transformer-based approach can generate high quality results for cell segmentation and tracking.

We also looked at the overall Cell-HOTA score, which integrates across all values of $\alpha$, and compared it to the Cell Tracking Challenge OP$_{CTB}$ score. With Cell-HOTA, we found that Trackastra had the highest overall score due to its strong tracking when coupled with the high-quality segmentation masks provided by EmbedTrack. When we provided Trackastra with segmentation masks generated from Cell-TRACTR, the Cell-HOTA score dropped from 91.16 to 89.29, demonstrating the reliance of Trackastra on high quality inputs from an outside source. Cell-TRACTR and EmbedTrack exhibited closely matched performance, while DeLTA performed slightly less well (Fig 4C). When using the OP$_{CTB}$ metric, Trackastra exhibited slightly higher overall performance across the four algorithms due to its strong TRA score and high SEG score. The TRA scores for all four models showed minimal variation, while there were differences in the SEG scores. This underscores the excessive weight given to the SEG score in OP$_{CTB}$. In addition, the TRA score offers limited insight into the model's actual detection and tracking performance (S1 Text). Overall, we found that the Cell-HOTA score offers an interpretable and balanced method for assessing model performance.

To understand the impact of permitting early and late divisions in Cell-HOTA, we evaluated the performance with and without flexible divisions (Fig 4D). All scores dropped significantly when flexible divisions were not allowed. Most notably, DivA consistently had the largest drop, demonstrating that all models struggled with pinpointing the exact frame a cell divides. It is clear that early and late divisions have a significant impact on the assessment of model performance. As the ground truth is often a subjective assessment of cell division time rather than the absolute truth, providing some leniency is likely to be critical for applications like bacterial cell tracking that exhibit many division events.

We also evaluated the time needed to process the test dataset in terms of frames per second (Fig 4E). The more frames a model can evaluate per second, the faster the model, thus larger values correspond to faster inference. Out of the models that performed segmentation and tracking, Cell-TRACTR had the fastest inference, with DeLTA performing second fastest. Although EmbedTrack had the slowest inference, its model utilized the least number of parameters and required the least amount of time to train. DeLTA utilized the most parameters and required the most time to train because it uses a separate model for segmentation and tracking whereas Cell-TRACTR and EmbedTrack only use one model. Since Trackastra only performs tracking, directly comparing inference speed and the total number of parameters is challenging. Therefore, we report these numbers for reference but note that they cannot be directly compared to the other models since they only include the time associated with tracking but not segmentation.

## Analyzing mammalian DeepCell microscopy movies

We next asked whether Cell-TRACTR would perform well on a very different cell segmentation and tracking task. For this, we considered data from the DynamicNuclearNet Tracking dataset from DeepCell [22]. In this dataset, five mammalian cells lines were cultured on 96-well plates. Nuclear staining was performed to assist with cell identification. In this two-dimensional dataset, the mammalian cells can grow in any direction, in contrast to bacterial growth in the mother machine

**A**

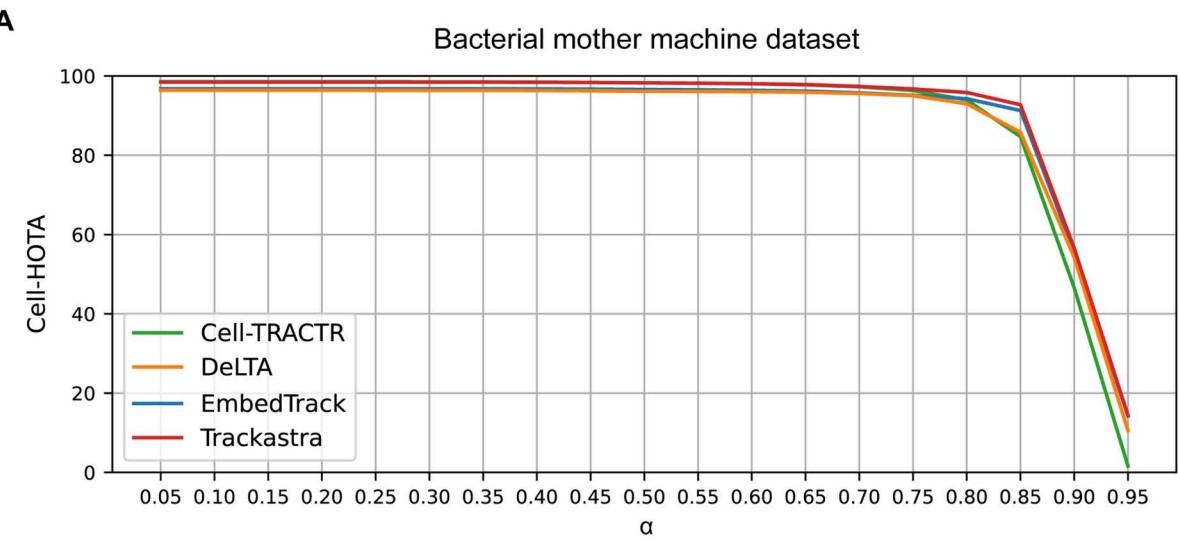

Bacterial mother machine dataset

**B**

Cell-HOTA$_{0.5}$

| | Cell-HOTA$_{0.5}$ | DetA$_{0.5}$ | AssA$_{0.5}$ | DivA$_{0.5}$ |
|---|---|---|---|---|
| Cell-TRACTR | **98.53** | 98.70 | **98.76** | **97.94** |
| DeLTA | 96.14 | 98.64 | 94.20 | 93.18 |
| EmbedTrack | 96.61 | **99.08** | 95.38 | 93.05 |
| Trackastra | 98.27 | 99.08 | 98.16 | 96.79 |

**C**

| | Cell-HOTA | | | | OP$_{CTB}$ | | |
|---|---|---|---|---|---|---|---|
| | Cell-HOTA | DetA | AssA | DivA | OP$_{CTB}$ | TRA | SEG |
| Cell-TRACTR | 89.63 | 89.03 | 92.55 | 88.89 | 0.934 | 0.985 | 0.883 |
| DeLTA | 88.72 | 90.58 | 89.81 | 85.20 | 0.938 | 0.982 | 0.895 |
| EmbedTrack | 89.63 | **91.43** | 91.88 | 85.50 | 0.943 | 0.984 | **0.902** |
| Trackastra | **91.16** | 91.43 | **94.57** | 88.90 | **0.944** | **0.986** | 0.902 |

**D**

| | Cell-HOTA | Cell-HOTA (No Flex) | DetA | DetA (No Flex) | AssA | AssA (No Flex) | DivA | DivA (No Flex) |
|---|---|---|---|---|---|---|---|---|
| Cell-TRACTR | 89.63 | 84.78 | 89.03 | 87.88 | 92.55 | 90.61 | 88.89 | 74.61 |
| DeLTA | 88.72 | 84.52 | 90.58 | 89.55 | 89.81 | 88.22 | 85.20 | 73.12 |
| EmbedTrack | 89.63 | 84.68 | 91.43 | 90.23 | 91.88 | 89.94 | 85.50 | 71.47 |
| Trackastra | 82.58 | 78.40 | 91.21 | 90.19 | 78.81 | 77.21 | 72.05 | 61.11 |

**E**

| | Inference (FPS) | Params | Training Time (hr) |
|---|---|---|---|
| Cell-TRACTR | 16.3 | 15.8M | 8.9 |
| DeLTA | 8.9 | 62.1M | 25.9 |
| EmbedTrack | 1.1 | 2.5M | 1.8 |
| Trackastra | 23.9* | 10.7M* | 8.2* |

*Only includes tracking

**Fig 4. Evaluation of model performance on the bacterial mother machine dataset.** (A) Cell-HOTA evaluated at various similarity thresholds α on the bacterial mother machine dataset, using data from the test set. Four different algorithms are evaluated: Cell-TRACTR, DeLTA, EmbedTrack, and Trackastra. (B) Cell-HOTA$_{0.5}$ results on the test set. (C) Overall results on the test set. Cell-HOTA has three sub-metrics: DetA, AssA, and DivA. OP$_{CTB}$

has 2 sub-metrics: SEG and TRA. (D) Cell-HOTA score with and without flexible divisions. (E) Comparison of model efficiency. FPS is the inference rate measured in frames per second, so higher values correspond to faster inference times. Params is the number of parameters in each model. M, million. Training time is the number of hours needed to train each model. Since Trackastra uses precomputed segmentation masks (from EmbedTrack), its performance does not reflect segmentation capability, unlike other models, which generate masks as a part of their pipeline. These values are denoted with a * because they only include tracking.

which is constrained to a single direction. The DeepCell dataset included images containing as few as 3 cells and as many as 281 cells in a single frame. In addition to DeLTA, EmbedTrack, and Trackastra, we also benchmarked against Caliban, which was specifically designed to perform well on the DeepCell dataset. Caliban utilizes a feature pyramid network [72] for segmentation and a long short-term memory model [73] for tracking. Both Caliban and Trackastra have demonstrated state-of-the-art results on the DynamicNuclearNet Tracking dataset. Following the Trackastra paper [41], we utilized the "General 2D" pretrained model and provided it the segmentation masks generated by Caliban.

Using the Cell-HOTA tracking metric, we evaluated the performance of Cell-TRACTR, DeLTA, EmbedTrack, Caliban, and Trackastra on the DeepCell dataset. Caliban demonstrated the highest overall performance, and Cell-TRACTR exhibited comparable performance for $\alpha < 0.50$ (Fig 5A). As before, we evaluated Cell-HOTA at $\alpha = 0.5$ (Fig 5B) and used the integrated Cell-HOTA score (Fig 5C). We found Caliban to have the highest overall Cell-HOTA$_{0.5}$, DetA$_{0.5}$ and DivA$_{0.5}$ score whereas Cell-TRACTR had the best AssA$_{0.5}$ score, though both models performed well across all these metrics (Fig 5B). For the overall Cell-HOTA score, we found that Caliban had the highest DetA and DivA scores whereas Trackastra had the highest AssA score (Fig 5C). The OP$_{CTB}$ metrics confirm EmbedTrack's overall quality in segmentation precision (SEG score), and Caliban achieved the top OP$_{CTB}$ TRA score (S1 Text). Overall, the results from the DeepCell dataset suggest that Cell-TRACTR exhibits detection, tracking, and division accuracy that are comparable to state-of-the-art algorithms (S2 Movie). This includes comparison to the Caliban algorithm, which was specifically tailored using the DeepCell dataset. As with the mother machine data, performance decreases with very stringent values of $\alpha$. Furthermore, these results also demonstrate that Cell-TRACTR can handle diverse cell segmentation and tracking tasks, as evidenced by the algorithm's strong performance on both bacterial and mammalian time-lapse microscopy data.

When analyzing inference rate, we found that Caliban had the fastest inference. Cell-TRACTR, EmbedTrack, and DeLTA had slower inference rates compared to the bacterial mother machine test data due to the larger image sizes and increased number of cells. EmbedTrack utilized the least number of parameters and required the least amount of training time. While Cell-TRACTR needed far fewer parameters than DeLTA, both required extensive training times. Although training times for Cell-TRACTR are currently long when image sizes are large, as we discuss below, recent improvements in computational efficiency for DETR-based algorithms could address this issue. Since we used pretrained models for Trackastra and Caliban, no training times are reported.

## Discussion

In this work, we developed a new transformer-based architecture, Cell-TRACTR, and showed that it can segment and track cells within time-lapse microscopy movies, exhibiting performance on par with current state-of-the-art software. This architecture, based on TrackFormer [44], can track cells in an end-to-end manner without any post-processing. The attention mechanism allows the transformer to make long-range connections, integrating information across distant parts of the image [37]. DETR-based models, such as those in [42,59], have emerged as state-of-the-art tracking models on the DanceTrack dataset, which features objects with irregular motion and uniform appearance, motivating our focus on them in this work. In practice, we found Cell-TRACTR to be well suited to address challenges associated with non-linear and rapid cell motion.

Cell-TRACTR uses track queries to take advantage of a cell's past history to make a coherent prediction in the subsequent frame. These track queries also allow for segmentation and tracking to be accomplished in one unified step which

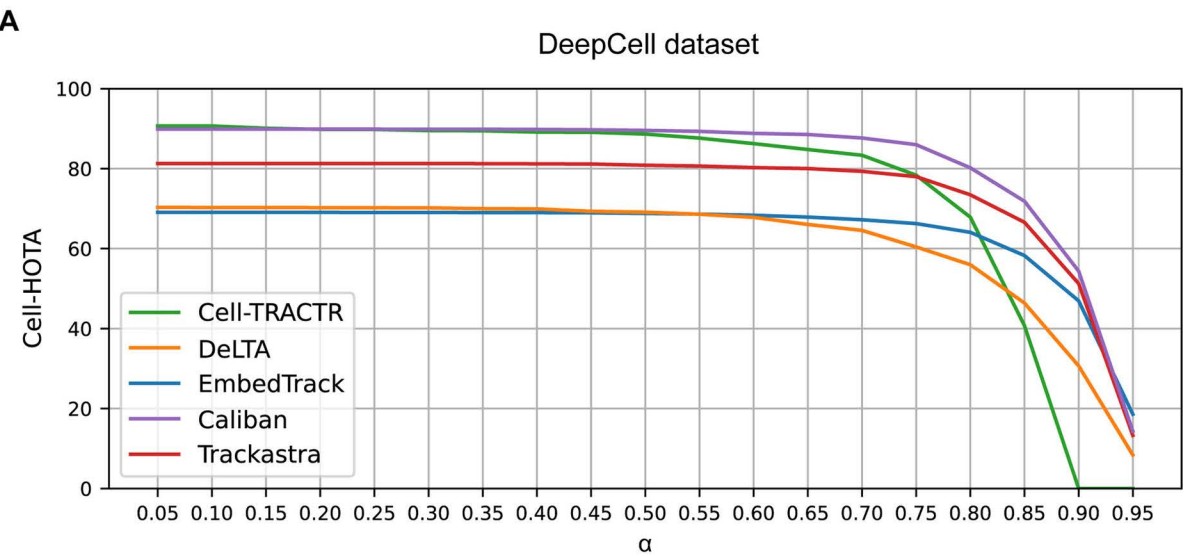

**A** DeepCell dataset

**B**

Cell-HOTA$_{0.5}$

| | Cell-HOTA$_{0.5}$ | DetA$_{0.5}$ | AssA$_{0.5}$ | DivA$_{0.5}$ |
|---|---|---|---|---|
| Cell-TRACTR | 88.65 | 96.04 | **95.81** | 69.90 |
| DeLTA | 69.11 | 92.81 | 89.97 | 29.43 |
| EmbedTrack | 68.79 | 91.71 | 91.13 | 29.22 |
| Caliban | **89.57** | **96.70** | 92.84 | **74.15** |
| Trackastra | 80.84 | 96.70 | 93.38 | 48.93 |

**C**

| | Cell-HOTA | DetA | AssA | DivA | OP$_{CTB}$ | TRA | SEG |
|---|---|---|---|---|---|---|---|
| Cell-TRACTR | 75.04 | 84.18 | 87.91 | 55.95 | 0.929 | 0.984 | 0.876 |
| DeLTA | 61.51 | 84.11 | 85.98 | 24.53 | 0.934 | 0.979 | 0.889 |
| EmbedTrack | 64.01 | 86.88 | 89.02 | 25.82 | **0.958** | 0.986 | **0.930** |
| Caliban | **82.07** | **91.10** | 89.71 | **64.03** | 0.951 | **0.989** | 0.913 |
| Trackastra | 74.47 | 91.12 | **90.26** | 42.91 | 0.950 | 0.988 | 0.912 |

**D**

| | Inference (FPS) | Params | Training Time (hr) |
|---|---|---|---|
| Cell-TRACTR | 0.50 | 28M | 84.0 |
| DeLTA | 0.04 | 62.1M | 64.2 |
| EmbedTrack | 1.00 | 2.5M | 6.8 |
| Caliban | 1.47 | 119.4M | N/A |
| Trackastra | 8.05* | 10.7M* | N/A |

*Only includes tracking

**Fig 5. Evaluation of model performance on the mammalian DeepCell dataset.** (A) Cell-HOTA evaluated at various similarity thresholds α on the mammalian DeepCell dataset, using data from the test set. Five different algorithms are evaluated: Cell-TRACTR, DeLTA, EmbedTrack, Caliban, and Trackastra. (B) Cell-HOTA$_{0.5}$ results on the test set. (C) Overall results on the test set. Cell-HOTA has three sub-metrics: DetA, AssA, and DivA. OP$_{CTB}$

has 2 sub-metrics: SEG and TRA. (D) Comparison of model efficiency. FPS is the inference rate measured in frames per second, so higher values correspond to faster inference times. Params is the number of parameters in each model. M, million. Training time is the number of hours needed to train each model. Since Trackastra uses precomputed segmentation masks (from Caliban), its performance does not reflect segmentation capability, unlike other models, which generate masks as a part of their pipeline. These values are denoted with a * because they only include tracking. Training times are not reported for Caliban and Trackastra because we used pretrained models.

may result in higher accuracy. Jointly addressing these tasks allows the model to leverage shared information between segmentation and tracking. Prior studies have shown that this can lead to better performance. For example, studies with TrackFormer showed that training a model jointly on tracking and segmentation improved the overall tracking performance compared to training a model on tracking alone [44]. In addition, MOTIP demonstrated that training a model jointly on object detection and tracking improved the HOTA score by a significant margin compared to first training a model on object detection and then using it as the foundation for training the tracking module separately [59]. Although other deep learning models, like EmbedTrack [24] and DistNet2D [23], can simultaneously segment and track cells, they typically require multiple decoders, whereas Cell-TRACTR uses a single decoder. This helps ensure that segmentation and tracking predictions are in sync across subsequent frames because the use of separate decoders for segmentation and tracking can lead to discrepancies in the predictions. A potential future advantage of this DETR-based architecture is that it could allow Cell-TRACTR to handle objects that are occluded or disappear. For example, Cell-TRACTR could discard track queries only if they are classified as 'no object' for multiple frames, as in TrackFormer [44]. To the best of our knowledge, Cell-TRACTR is the first algorithm to simultaneously segment and track cells with a single decoder while requiring no post-processing.

To assess algorithm performance, we extended the widely-used HOTA metric to incorporate cell division, creating the Cell-HOTA metric. We also calculated performance using the traditional Cell Tracking Challenge $OP_{CTB}$ metric as a point of comparison. Cell-HOTA can be broken down into detection, association, and division accuracy, all of which are calculated across a range of similarity thresholds, $\alpha$. We found that Cell-TRACTR performed well on association and division accuracy while maintaining solid detection capabilities. However, Cell-TRACTR had reduced segmentation precision compared to other algorithms at high similarity thresholds. It is important to note that Cell-TRACTR may not be the optimal choice for datasets where segmentation precision is the top priority (i.e., cases where very high values of the similarity threshold $\alpha$ are required). Overall, the Cell-HOTA metric enables the assessment of detection, association, and division, offering crucial insights for evaluating algorithm performance in cell tracking applications.

We found that Cell-TRACTR delivered strong performance on both a bacterial mother machine dataset and a mammalian dataset from DeepCell. However, we did encounter computational limits when working with large two-dimensional datasets. For example, training times for two-dimensional images containing hundreds of cells in 584x600 pixel images exceeded a week, making hyperparameter optimization infeasible, leading us to conclude that this goal was impractical with the current version of Cell-TRACTR. A well-documented limitation of transformers is their computational complexity, a challenge that can be pronounced with microscopy data, where image sizes can be large, and images contain many cells at low resolution. The need to process large images leads to large multi-scale feature maps, while the low resolution of the cells necessitates an increase in the number of multi-scale feature maps, making this a computationally demanding task. However, there is recent promising work focused on making DETR-based models more computationally efficient, which offers future potential for this area. Examples include RT-DETRv3 [74], DEIM [75], and Efficient DETR [52], which increase computational efficiency, while SOF-DETR [76] and work by Huang *et al*. [77] offer approaches that are better at handling small objects. These new models mainly focus on engineering the encoder to be more efficient. Building upon the momentum for increased computational efficiency, DEFA [78] proposed a more efficient form of deformable attention that significantly reduces memory usage and training time. Departing from the tracking-by-query paradigm, MOTIP [59] utilizes a DETR-based model to generate object embeddings for each frame which are used to address an ID prediction problem similar to Trackastra [41]. Specifically focusing on segmentation, CelloType [40], which is based on Mask DINO

[53], has demonstrated top performance compared to other state-of-the-art cell segmentation models. Similarly, it would be interesting to explore a two-stage training approach for Cell-TRACTR, first pretraining the model on cell segmentation and then fine-tuning it for cell tracking. This training strategy could improve computational efficiency by allowing parts of the model to be frozen during fine-tuning. For instance, when adapting the model to tracking, the backbone weights could be frozen, reducing computational overhead. Given the recent focus on improving the computational efficiency of DETR-like algorithms, we anticipate that future advances in this space will likely address some of the limitations we encountered. Integrating these improvements into Cell-TRACTR could enable cell segmentation and tracking of larger and increasingly complex time-lapse images.

Another computational challenge is that DETR-based models that perform segmentation require a large amount of memory, as each output embedding produces a segmentation mask of the whole image. This is computationally demanding, for example when there are 250 cells in an image, 250 full-sized segmentation masks will be produced. When working with large two-dimensional images, we sometimes encountered CUDA out-of-memory errors during backpropagation for the segmentation masks. Future efforts could focus on only backpropagating the loss through sampled regions within the segmentation mask to reduce the computational load. Recently, Frozen Mask-DETR [71] developed a computationally efficient way to segment images with DETR. The model uses DINO-DETR [50] to predict bounding boxes, which are used as regions of interest that are segmented by an Instance Mask Decoder. This method significantly reduces the computational load by only segmenting the area within the bounding box while also outperforming Mask-DINO [53] in instance segmentation.

Similar to our work, Trackastra developed a computationally efficient way to track cells utilizing transformers [41]. Their method consists of feeding object positions and shallow object features to a transformer and outputting an association matrix. Shallow object features include characteristics such as cell area and cell shape. The association matrix details how cells are related between two adjacent frames. Different from Trackastra, our work utilizes a tracking-by-query paradigm, can segment cells in addition to tracking cells, and requires no post-processing. While Trackastra has developed a Napari plugin, a future direction for Cell-TRACTR would be to integrate our algorithm into a user-friendly workflow to improve accessibility. Trackastra is a nice complement to our work and is a lightweight transformer-based model for cell tracking.

Cell-TRACTR currently processes one frame at a time, whereas some other cell tracking algorithms analyze multiple frames at a time to incorporate temporal information into their models [22,23,28] and this is a potential extension of this work. Specifically, in Cell-TRACTR, track queries only attend to image features in the current frame of interest, however, they could attend to the previous and future frames for additional context. This could be achieved by stacking multi-scale features from a sequence of frames and feeding them to the encoder, as in TrackFormer [44]. However, the feasibility of this approach is contingent on significant improvements in the computational efficiency of the encoder and decoder. Another extension of this work could include using the output embeddings for other auxiliary tasks like performing classification based on cell morphology [40]. For such tasks, the weights of Cell-TRACTR could be frozen to ensure the segmentation and tracking performance remain high while minimizing the computational load.

The Cell-HOTA metric could also be extended in several ways. We designed Cell-HOTA to work well on time-lapse images of cells that divide frequently. As implemented, Cell-HOTA weighs DivA and AssA equally, which places a high emphasis on division accuracy. For other use cases where cell division is less common, it may be more appropriate to adjust the metric to de-emphasize DivA relative to the other accuracy values. In addition, a current limitation of Cell-HOTA is that it only accepts flexible divisions that are one frame early or late. This could be modified to allow early or late divisions that occur two or more frames away from the ground truth.

While this manuscript was under review, Kaiser et al. [79] introduced a novel cell tracking metric called CHOTA, which is similar to Cell-HOTA. CHOTA also extends the HOTA framework by leveraging the DetA and AssA sub-metrics. However, unlike Cell-HOTA, which incorporates a Division Accuracy (DivA) sub-metric to explicitly evaluate the accuracy of division predictions, CHOTA shifts the focus to lineage-level trajectory analysis. In Cell-HOTA, AssA is calculated for each

individual cell, regardless of whether it divides. In contrast, CHOTA evaluates AssA at the lineage level, measuring association not just for a single cell, but also for its progeny. For instance, if a cell divides, CHOTA measures AssA for the parent cell as well as its two daughter cells, providing a more holistic view of lineage continuity. While CHOTA incorporates lineage information in the metric, this approach can decrease interpretability. Cell-HOTA, on the other hand, provides a distinct assessment of model performance in predicting cell division events, making it easier to diagnose errors. Additionally, Cell-HOTA places a greater emphasis on division accuracy and uniquely accounts for early and late divisions. Overall, Cell-HOTA is better suited for applications where interpretable feedback is prioritized, while CHOTA is more appropriate for applications where lineage continuity takes precedence. Both metrics offer marked improvements over the traditional $OP_{CTB}$ metric for balancing segmentation and tracking accuracy.

In summary, we introduce Cell-TRACTR, a novel deep learning model that can segment and track cells across subsequent frames in time-lapse images. Our model uses track queries, which leverage a cell's history, to simultaneously segment and track cells. The self-attention mechanism in the decoder plays a crucial role in eliminating duplicate predictions and differentiating between object and track queries. We also introduced a new cell tracking metric, Cell-HOTA, and demonstrated that Cell-TRACTR performs favorably compared to other state-of-the-art cell tracking algorithms and is particularly well-suited to association and division tasks. As transformers and DETR-based models continue to progress, we anticipate that Cell-TRACTR can be adapted to increase efficiency, enabling it to handle predictions using larger images and those crowded with cells.

## Methods

### Training Cell-TRACTR

For the bacterial mother machine dataset, we trained Cell-TRACTR for 12 epochs with a batch size of 2. The number of object queries was set to 30. The transformer consisted of 4 encoder layers and 4 decoder layers. The target size was 256x32 pixels. Images had to be resized to fit the target size. All configurations and hyperparameters are detailed in train_moma.yaml file within the cfgs folder on the GitLab repository. A V100 GPU was utilized.

For the DeepCell dataset, we trained Cell-TRACTR for 24 epochs with a batch size of 1. The number of object queries was set to 400. The transformer consisted of 4 encoder layers and 4 decoder layers. The target size was 584x600 pixels. Images were not resized since they were already the correct target size. All configurations and hyperparameters are detailed in train_DynamicNuclearNet-tracking-v1_0.yaml file within the cfgs folder on the GitLab repository. We used the A100 GPU to alleviate out-of-memory issues (S1 Text).

Model checkpoints are available on Zenodo (https://zenodo.org/records/14509424).

Next, we detail important parameters used to train Cell-TRACTR for both datasets. We used a learning rate of 0.0002 for the encoder-decoder and a learning rate of 0.00002 for the CNN backbone. Both learning rates were dropped by a factor of 10 at epoch 12 and 20 for the models trained on the bacterial mother machine and DeepCell datasets, respectively. We used AdamW [80] as the optimizer with a weight decay of 0.0001. We adopted ResNet-18 as the backbone CNN for the bacterial mother machine dataset and ResNet-50 as the backbone CNN for the DeepCell dataset and fed the last 3 feature maps to the encoder. The first decoder layer was used for object detection and subsequent layers were used for tracking (S5 Fig and S1 Text).

We used a clip size of up to 3 frames to train the model. The model is either fed 1, 2, or 3 frames. When 1 frame is fed to the model, only object detection is performed. When 2 frames are fed to the model, object detection is performed in the first frame and tracking is performed on the subsequent frame. When 3 frames are fed to the model, object detection is performed in the first frame and tracking is performed on the two subsequent frames. We do not exclusively feed the model 3 frames because it needs to be trained explicitly on object detection as the loss is only calculated for the final frame within the clip. For example, when 3 frames are fed to the model, backpropagation will not affect false positives predicted during object detection in the first frame. It is important that the model learns to track for at least 3 sequential

frames because the model needs to track cells that have just divided. This is a unique task for DETR-based tracking models, as cells that have just divided contain different positional embeddings but the same content embeddings (S6 Fig).

Following the approach used in TrackFormer [44], we utilize false negatives and false positives during training to improve the robustness of the model. False negatives are incorporated to teach the model to identify cells that were previously undetected due to errors. We implemented conditions where there is a probability of 0.5 that at least one of the track queries will be dropped, with a maximum of 20% of the track queries potentially being dropped. When a track query is dropped, an object query is forced to detect the cell. False positives are also incorporated to ensure that the model can ignore erroneous predictions made in previous frames. In our implementation, there is a probability of 0.5 that at least one false positive will be added, with a maximum of 20% of the total number of track queries added as false positives. False positives are generated as noisy track queries. To generate the positional embedding for the false positive, random noise is added to the bounding box of a track query with $\lambda_1 = 0.2$ and $\lambda_2 = 0.1$ (S1 Text). To generate the content embedding for the false positive, random noise drawn from a normal distribution with a mean of 0 and a standard deviation of 0.25 is added to the content embedding of the respective track query. False positives and false negatives were only incorporated in the final frame of the clip when the clip size was greater than 1. Compared to TrackFormer [44], we adjusted the frequency of false positives and false negatives and the generation of the positional and content embeddings for the false positives to simplify the code.

## Loss function

The loss function is a culmination of losses, including the class label loss, bounding box loss, and segmentation mask loss. The class label loss is calculated using focal loss with class weighting factor $\alpha = 0.25$ and focusing parameter $\gamma = 2$. The bounding box loss is a linear combination of the L1 loss and generalized IOU loss. The segmentation mask loss is a linear combination of the Dice Loss and binary cross entropy. To account for cell divisions, we average the loss across the two divided cells. Therefore, a cell division would have equal weight compared to a single cell. The loss functions were taken directly from DETR [46] and modified to handle cell divisions.

## Data augmentations

To help generalize our model, we applied three data augmentation techniques during training: random Gaussian blur, random Gaussian noise, and random illumination adjustments. All three techniques were applied with a probability of 0.4 for each augmentation for a sequence of frames. These were chosen to reflect imaging conditions seen in microscopy data. These data augmentations were all implemented using the approach described in [10,20].

## Training DeLTA

For the bacterial mother machine dataset, we trained the segmentation and tracking models for 12 epochs with a batch size of 1. The target size was 256x32 pixels. The "mother machine" configuration in DeLTA was utilized. All default data augmentations were included.

For the DeepCell dataset, we trained the segmentation and tracking model for 24 epochs with a batch size of 1, incorporating a patience parameter of 5 epochs. The target size was 256x256 pixels. The "2D" configuration in DeLTA was used, in addition to all default data augmentations.

Additionally, we modified the code to directly train on Cell Tracking Challenge formatted datasets.

## Training EmbedTrack

For the bacterial mother machine dataset, we trained EmbedTrack for 12 epochs with a batch size of 16. Although cropping images is the default for EmbedTrack, the small size of the mother machine images made this step unnecessary as cropping is used to circumvent out-of-memory errors. Instead, we adjusted the code to resize all images to a uniform input

size of 256x32 pixels. We removed the random rotation data augmentations from EmbedTrack, as they were incompatible with a crop size that was not a square. A V100 GPU was used for training.

For the DeepCell dataset, we trained EmbedTrack for 24 epochs with a batch size of 4. The batch size was reduced from the default size of 16 to prevent out-of-memory errors. The images were cropped to 256x256 pixels, which is the default crop size for EmbedTrack. Since the images in this dataset were cropped during training and inference, there was no need for additional resizing. All default data augmentations from EmbedTrack were utilized. A V100 GPU was used for training.

All models were trained for 12 epochs on the bacterial mother machine dataset and 24 epochs on the mammalian DeepCell dataset. Each epoch involved iterating through every image in the dataset once, ensuring equal training time for each model. We used the default configurations and training parameters when applicable.

### Training Trackastra

For the bacterial mother machine dataset, we trained Trackastra using the example_config.yaml file provided on the GitHub page (https://github.com/weigertlab/trackastra). For the DeepCell dataset, we used the code and pretrained weights provided by the Weigert lab. Specifically, we used the "general_2d" model and the ILP based linking as that was shown to perform better than the greedy linking approach [41].

### Training Caliban

We used the code and pretrained weights provided by the Van Valen lab (https://github.com/vanvalenlab/Caliban-2024_Schwartz_et_al).

### HOTA metric overview

To calculate the HOTA score, we used the approach from [62], which we summarize here for completeness. We first used the Hungarian algorithm [81] to match the tracker cells with the ground truth cells that generated the highest HOTA score. DetA is measured as the Jaccard Similarity where a match is considered a true positive (TP) if IOU > α (Equation 1). False positives (FP) occur when a tracker cell does not match to any ground truth cells and false negatives (FN) occur when a ground truth cell does not match to any tracker cells. A match with an IOU below the threshold α is considered a FP for the tracker cell and a FN for the ground truth cell. To instill localization accuracy into the detection accuracy, DetAα is measured at multiple thresholds of α. In other words, the detection accuracy depends upon the value of the threshold α.

$$DetA_{\alpha} = \frac{TP}{TP + FP + FN}$$
(1)

To calculate AssA, for all the true positives detected at each α, the number of matches for each tracker and ground truth cell are enumerated (Equation 2). A false positive association (FPA) occurs when the tracker cell matches to a different ground truth cell. A false negative association (FNA) occurs when the ground truth cell matches to a different tracker cell. A(c) is the association score for a single detected true positive where c is a detected true positive. AssAα calculates the association score over all detected true positives for a given value of α (Equation 3).

$$A(c) = \frac{TPA(c)}{TPA(c) + FPA(c) + FNA(c)}$$
(2)

$$AssA_{\alpha} = \frac{1}{|TP|} \sum_{c \in \{TP\}} A(c)$$
(3)

HOTAα is calculated by taking the geometric mean of DetAα and AssAα (Equation 4). This is typically measured at values of α ranging from 0.05 to 0.95 in increments of 0.05.

$$HOTA_\alpha = \sqrt{DetA_\alpha * AssA_\alpha} \tag{4}$$

The final HOTA score is calculated by taking the average Cell-HOTAα score across all α values (Equation 5).

$$HOTA = \int_0^1 HOTA_\alpha \, d\alpha \approx \frac{1}{19} \sum_{\alpha \in \{ \substack{0.05, \ 0.1, \ \ldots \\ 0.9, \ 0.95} \}} HOTA_\alpha \tag{5}$$

## Cell-HOTA metric

To allow Cell-HOTA to handle cell division, we added a new sub-metric called division accuracy, DivA. DivAα is measured as the Jaccard Similarity across all cell divisions at a similarity threshold α (Equation 6). DivAα is calculated using the same approach as DetAα

$$DivA_\alpha = \frac{TPD}{TPD + FPD + FND} \tag{6}$$

We define AssDivAα as the geometric mean of AssAα and DivAα at a similarity threshold α (Equation 7). Association and division accuracy are equally weighted to emphasize the importance of division accuracy.

$$AssDivA_\alpha = \sqrt{AssA_\alpha * DivA_\alpha} \tag{7}$$

Cell-HOTAα is the geometric mean of DetAα and AssDivAα at a similarity threshold α (Equation 8). This calculation is the same as HOTAα except AssDivAα replaces AssAα.

$$\text{Cell-}HOTA_\alpha = \sqrt{DetA_\alpha * AssDivA_\alpha} \tag{8}$$

Cell-HOTA is a measured by taking the average Cell-HOTAα score across all values of α (Equation 9). This is calculated using the same approach as with HOTA.

$$\text{Cell-}HOTA = \int_0^1 \text{Cell-}HOTA_\alpha \, d\alpha \approx \frac{1}{19} \sum_{\alpha \in \{ \substack{0.05, \ 0.1, \ \ldots \\ 0.9, \ 0.95} \}} \text{Cell-}HOTA_\alpha \tag{9}$$

## Cell-HOTA and flexible divisions

A flexible early division occurs when a tracker cell divides at time, for example, $t_1$ and the matched ground truth cell divides in the next frame, $t_2$ in this example (S7 Fig). To check for flexible early divisions at time $t_1$, we iterate through all ground truth cells that divide at time $t_2$ and match to a tracker cell that divides at time $t_1$. Next, we check that the tracker and ground truth parent cells at time $t_0$ match and the tracker and ground truth daughter cells match at time $t_2$. Similarly, a flexible late division occurs when a ground truth cell divides at time $t_1$ and the matched tracker cell divides in the next frame at time $t_2$. To check for flexible late divisions at time $t_1$, we iterate through all tracker cells at time $t_1$ and check if

any divide at time $t_2$ and match to a ground truth cell that divides at time $t_1$. Next, we check that the tracker and ground truth parent cells at time $t_0$ match and the tracker and ground truth daughter cells match at time $t_2$. For both flexible early and late divisions, we check that the average IOU between the daughter cells at time $t_2$ is greater than the similarity threshold α.

When a flexible division is detected, the DetAα score is adjusted as if the tracker cell divided in the same frame as the ground truth cell. For example, when there is a flexible early division, the IOU is calculated between two tracker cells and one ground truth cell where the two tracker cells are treated as one object (S7 Fig). When a flexible late division occurs, the IOU is calculated between the two ground truth cells and one tracker cell where the two ground truth cells are treated as one. This IOU value is counted twice since there are two ground truth cells. This is done to ensure each ground truth cell has the same weight and the score remains the same whether a tracker division was predicted at the correct time or a frame early or late.

As with DetAα, we adjust the AssAα score to match the ground truth for flexible divisions. When a flexible early division occurs, a match is added between the ground truth cell and parent of the tracker cells and a match is removed from the ground truth cell and one of the daughter tracker cells. When a flexible late division occurs, a match is added between the two ground truth cells and the two divided tracker cells in the subsequent frame and a match is removed between one of the ground truth cells and the tracker cell in the current frame.

## Supporting information

**S1 Text. Includes supplementary text, methods, and references.**
(PDF)

**S1 Movie. Time-lapse microscopy movie from the bacterial mother machine test set processed by Cell-TRACTR.** Different colors represent unique cells being tracked. Black arrows indicate cell division. The temporal resolution of the movie is one frame every five minutes. Both the bounding boxes and segmentation masks are displayed for visualization purposes. However, during inference, only the segmentation masks are used.
(MP4)

**S2 Movie. Time-lapse microscopy movie from the mammalian DeepCell test set processed by Cell-TRACTR.** Different colors represent unique cells being tracked. Black arrows indicate cell division. The temporal resolution of the movie is one frame every five minutes. Both the bounding boxes and segmentation masks are displayed for visualization purposes. However, during inference, only the segmentation masks are used.
(MP4)

**S3 Movie. Time-lapse microscopy movie from the bacterial mother machine test set processed by Cell-TRACTR and evaluated with the Cell Tracking Challenge (CTC) OP$_{CTB}$ metrics.** From left to right, the first column shows the raw images, the second column shows the ground truths, and the third column shows the predictions. Distinct colors represent unique cells being tracked over time. The right six columns show the operations used to transform the predicted graph into the reference graph provided by the ground truth. The operations include delete vertex (FP), add vertex (FN), split vertex (NS), add edge (EA), delete edge (ED), and alter the edge semantics (EC). When one of these operations is applied, the cell is shown in red.
(MP4)

**S1 Fig. The encoder employs deformable self-attention to transform multi-scale features into encoded multi-scale features.** Query selection is used to derive the object queries from these encoded multi-scale features. Each encoded image feature undergoes classification, and the top-K features are selected as the content embeddings for the object queries. Additionally, bounding boxes are extracted from the top-K segmentation masks and are subsequently

converted into positional embeddings. Following this, the object queries perform self-attention and deformable cross-attention with the encoded multi-scale features to produce the output embeddings. MLP, multi-layer perceptron. (TIF)

**S2 Fig.  Example illustrating how object and track queries interact using an attention map processed by Cell-TRACTR across two subsequent frames.** The schematic displays the actual output alongside the output the from top matching object queries. Extracted from the self-attention mechanism in the decoder, the attention map highlights the strength of interactions among the queries. Lighter colors indicate stronger interactions. The track queries tend to focus on themselves, while the object queries attend to the track query that corresponds most closely in location. (TIF)

**S3 Fig.  Depiction of how predictions are made from the output embeddings.** The largest encoded multi-scale feature map is resized and combined with the original multi-scale features to form the pixel embedding map. The dot product of the output embeddings and the pixel embedding map produces the segmentation masks. A linear layer is used to predict the class label and a multi-layer perceptron (MLP) is used to generate the bounding box. For output embeddings classified as "no object", the bounding boxes and segmentation masks are ignored. (TIF)

**S4 Fig.  Example showing how $OP_{CTB}$ can reward a tracker for predicting no division rather than predicting a division one frame too early or late.** Acyclic oriented graph matching (AGOM) is the weighted sum of the operations needed to transform the tracker graph into the reference graph. A lower AGOM score is better and a higher TRA score is better. (TIF)

**S5 Fig.  Cells are initially detected in frame $t_0$ and then tracked to frame $t_1$.** Object queries are processed in the first layer of the decoder for object detection. In the subsequent decoder layers, track queries are concatenated with the object queries to perform tracking. (TIF)

**S6 Fig.  Schematic demonstrating how Cell-TRACTR tracks cells post-division.** The dark blue cell at time $t_1$ is predicted to divide at time $t_2$. The content embedding is duplicated for both divided cells. The positional embeddings are generated from each divided cell's segmentation mask. The content embeddings are shown as squares and the positional embeddings are circles. (TIF)

**S7 Fig.  Schematic demonstrating how Cell-HOTA handles early and late divisions.** For early divisions, the ground truth cell is originally matched with one of the tracker cells at time $t_1$. The red line indicates the matched cells, and the dotted blue box shows how the IOU is computed. To account for an early division, the IOU is computed between the ground truth cell in $t_1$ and both tracker cells in $t_1$. In addition, the temporal association for the ground truth cell in $t_1$ is altered to match the tracker cell in $t_0$. Similarly, for late divisions, the tracker cell is originally matched to one of the ground truth cells at time $t_1$. To account for a late division, the IOU is computed between the tracker cell in $t_1$ and both ground truth cells in $t_1$. The temporal associations are added between the ground truth cells in $t_1$ and the tracker cells in $t_2$. Note that for clarity this schematic only shows changes that are affected by the early or late division. (TIF)

**S8 Fig.  (A) Example illustrating the correlation between the final prediction of an object query and where the object query is derived from within the multi-scale features.** Colored pixels represent multi-scale features that were classified as "cell" by the decoder. Black pixels represent multi-scale feature pixels that were classified as "no object" by

the decoder. The colored pixels are overlaid onto the phase contrast image for visualization purposes. (B) A more complex example illustrating the correlation between the multi-scale features and object queries.
(TIF)

**S9 Fig. Heatmap illustrating which multi-scale feature pixels were classified as "cell" by query selection.** This data was collected by processing the full test set containing 29 movies from the bacterial mother machine dataset with Cell-TRACTR. Note that for these movies, the model preferentially uses pixels from the lowest resolution multi-scale features. The black pixel at the top corresponds to the top of the mother machine image where there are not usually cells.
(TIF)

**S10 Fig. Heatmap illustrating which multi-scale feature pixels were classified as "cell" by query selection.** This data was collected by processing the full test set for the DeepCell dataset. Note that for these movies, Cell-TRACTR preferentially uses pixels from the highest resolution multi-scale features.
(TIF)

**S11 Fig. Diagram illustrating how the iterative bounding box refinement method works with cell divisions.** When tracking the cell from $t_0$ to $t_1$, the bounding box from $t_0$ serves as the initial bounding box that is iteratively refined three times into a bounding box that locates the cell in $t_1$. The second bounding box is ignored here since cell division is not predicted at time $t_1$. This method consists of predicting a relative offset with respect to the previous bounding box, represented by a change in cell location and size ($\Delta y$, $\Delta x$, $\Delta h$, $\Delta w$). This offset is added to the previous bounding box to get the new prediction. After the first layer in the decoder, the predicted relative offset is added to the initial bounding box in $t_1$. This new bounding box serves as the reference point for the next layer. The reference point indicates where the query should focus within the encoded multi-scale features. In the two subsequent layers, the predicted relative offset is added to the previous bounding box to get the final bounding box location. At time $t_2$, the model predicts a cell division resulting in both relative offsets being utilized. After the first layer in the decoder, the predicted relative offsets are added to the same initial bounding box in $t_2$. In the two subsequent layers, the two predicted relative offsets are added to their respective previous bounding box. We generate a new bounding box around both divided cells that serves as the new reference point. This only occurs when both class labels for a track query are greater than 0.5.
(TIF)

**S12 Fig. Cell-TRACTR detects cells in the first frame at time t 0.** Object queries predicted to be a "cell" are converted to track queries. For Track Group Denoising (TGD), random noise is added to the track queries to get noised track queries. For Query Denoising (QD), random noise is added to the ground truth bounding boxes to generate noised track queries. All queries are processed together by the decoder. Attention masks are used to prevent information leakage between the object and track queries, the noised track queries generated from TGD, and the noised track queries generated from QD. The gray boxes indicate information being blocked between sets of queries while colored boxes indicate the flow of information.
(TIF)

**S13 Fig. Effect of segmentation mask resolution on CUDA memory usage and performance metrics.** (A) Memory usage as a function of queries when Cell-TRACTR produces a segmentation mask with the same resolution as the original image. (B) 1/4 Cell-TRACTR produces a segmentation mask with one-fourth the resolution of the input image.
(TIF)

## Acknowledgments

We thank Jean-Baptiste Lugagne and all members of the Dunlop Lab for helpful discussions.

## Author contributions

**Conceptualization:** Owen M. O'Connor.

**Investigation:** Owen M. O'Connor.

**Software:** Owen M. O'Connor.

**Supervision:** Mary J. Dunlop.

**Writing – original draft:** Owen M. O'Connor, Mary J. Dunlop.

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
