## [Decision Letter · Decision Letter 0]

24 Oct 2024

Dear Dr Dunlop,

Thank you very much for submitting your manuscript "Cell-TRACTR: A transformer-based model for end-to-end segmentation and tracking of cells" for consideration at PLOS Computational Biology.

As with all papers reviewed by the journal, your manuscript was reviewed by members of the editorial board and by several independent reviewers. In light of the reviews (below this email), we would like to invite the resubmission of a significantly-revised version that takes into account the reviewers' comments.

Please give special attention to the reviewers' concerns about reproducibility and comparison with existing methods, especially Trackastra and CALIBAN.

We cannot make any decision about publication until we have seen the revised manuscript and your response to the reviewers' comments. Your revised manuscript is also likely to be sent to reviewers for further evaluation.

Sincerely,

Virginie Uhlmann

Academic Editor

PLOS Computational Biology

Jason Haugh

Section Editor

PLOS Computational Biology

Please give special attention to the reviewers' concerns about reproducibility and comparison with existing methods, especially Trackastra and CALIBAN.

Reviewer's Responses to Questions

**Comments to the Authors:**

Reviewer #1: Cell-TRACTR is a transformer-based deep learning algorithm that combines a convolutional neural network for visual feature extraction with a transformer encoder-decoder architecture, inspired by the DETR model from Carion et al. (2020). This method is the first to apply DETR to biological image tracking, but it has some limitations:

Major points

1. Limited Improvement: in comparison to the non-transformer-based methods, the improvement by Cell-TRACTR needs more investigation. The presented method shows little improvement with the first dataset and inconsistent improvement with the second dataset. Especially with the second dataset, the real performance gain is in the cell division accuracy component (DivA) of Cell-HOTA. The reason for this is not clear. This does not necessarily mean that the presented method is ineffective, but it will be more useful for the audience if the authors present more specific use cases where the DivA component would make a crucial difference. Adding such datasets or more detailed analysis on the DeepCell Dataset, such as ablation studies, might prove helpful.

2. Scalability Concerns: Due to DETR’s bipartite matching, the number of detectable/trackable objects is limited by the specified number of queries (class label, bounding box, and segmentation mask). How does Cell-TRACTR handle images with numerous objects or significant variance in object numbers between samples? Additionally, insights into how computational cost scales with the expected maximum number of objects would be valuable.

Minor points

1. Lack of Comparison with Transformer-Based Models: The method is not compared to other transformer-based models. It would be useful to evaluate its performance against models like Trackastra, followed by cell-segmentation methods, to assess the impact of the transformer architecture.

2. Comparison with Common Workflows: Comparing Cell-TRACTR with popular workflows, such as commonly used configurations from Trackmate software, would be beneficial for biologists.

3. Figure 2 could be more concise by reducing the number of repetitive objects.

Reviewer #2: The review is uploaded as an attachment (202409_Review_PLOS_Comp_Biol.md).

Reviewer #3: Major comments:

1. It's difficult to assess the proposed method's performance in relation to the existing literature as a whole because it was evaluated in only two datasets against two algorithms (one of which is from the same authors). This is especially true since they propose a new metric, which is indeed needed. However, it reduces the comparability to existing work despite them reporting the CTC score, which, as they said, is problematic. We recommend comparing it with CALIBAN [22], which produced the DeepCell dataset used in this paper, and Trackastra [40], which is also a transformer-based method for tracking and the current state-of-the-art in the DeepCell dataset. Both of these methods have their code and weights publicly available.

2. A concurrent work, Cell-specific Higher Order Tracking Accuracy (CHOTA)[1*, 2*], which extends HOTA for cell tracking, was recently proposed. It doesn't seem to be the same metric as yours, so it would be interesting to highlight the differences between the two from the theoretical and experimental point of view (e.g., how they measure different things).

3. It's not clear how to obtain the network weights used in the experiments. The weights are as important as making the code available, especially in this case where server-grade GPU (V100/A100) are needed to train the models. Otherwise, the reproducibility and application of the proposed method are greatly reduced.

4. This method's application seems a bit limited because it requires an extensive amount of GPU RAM, an A100 for 584x600 pixel images with a batch size of 1, which are small images for today's standards. Plus, it seems to require a considerable amount of training data. How does it perform in videos from the Cell Tracking Challenge which are often more challenging than the DeepCell dataset?

Minor comments:

5. Omnipose appears twice in the references [8, 32]

6. From section "Training EmbedTrack." How are (horizontal or vertical) flip augmentations incompatible with a fixed crop size?

[1*] Kaiser, Timo, Vladimir Ulman, and Bodo Rosenhahn. "CHOTA: A Higher Order Accuracy Metric for Cell Tracking." arXiv preprint arXiv:2408.11571 (2024).

[2*] http://www.bioimagecomputing.com/program/selected-contributions/

**Have the authors made all data and (if applicable) computational code underlying the findings in their manuscript fully available?**

Reviewer #1: **No: ** The code repository is not visible in the Data and Code Availability section.

Reviewer #2: Yes

Reviewer #3: Yes

PLOS authors have the option to publish the peer review history of their article (what does this mean? ). If published, this will include your full peer review and any attached files.

**Do you want your identity to be public for this peer review?** For information about this choice, including consent withdrawal, please see our Privacy Policy .

Reviewer #1: No

Reviewer #2: No

Reviewer #3: No
---

## [Decision Letter · Decision Letter 1]

12 Feb 2025

PCOMPBIOL-D-24-01171R1

Cell-TRACTR: A transformer-based model for end-to-end segmentation and tracking of cells

PLOS Computational Biology

Dear Dr. Dunlop,

Thank you for submitting your manuscript to PLOS Computational Biology. After careful consideration, we feel that it has merit but does not fully meet PLOS Computational Biology's publication criteria as it currently stands. Therefore, we invite you to submit a revised version of the manuscript that addresses the points raised during the review process.

As you can read from the reviews, Reviewers 1 and 2 are satisfied with the revised manuscript (though note the suggestions for code changes raised by Rev. 2), Reviewer 3 has expressed lingering concern about the ability to generalize the method -- in a computationally efficient manner -- to other datasets; this affects the prospects for widespread adoption in their view.

Please submit your revised manuscript within 30 days Apr 14 2025 11:59PM. If you will need more time than this to complete your revisions, please reply to this message or contact the journal office at ploscompbiol@plos.org. Please include the following items when submitting your revised manuscript:

We look forward to receiving your revised manuscript.

Kind regards,

Jason M. Haugh

Section Editor

PLOS Computational Biology

**Journal Requirements:**

We have noticed that you have a list of Supporting Information legends in your manuscript for **S1 Movie, S2 Movie, S3 Movie** . However, there are no corresponding files uploaded to the submission. Please upload them as separate files with the item type 'Supporting Information'.

**Reviewers' comments:**

Reviewer's Responses to Questions

**Comments to the Authors:**

Reviewer #1: The authors have thoroughly addressed all of my concerns and those of the other reviewers.

Reviewer #2: My concerns regarding the manuscript have been addressed by the authors.

Also, following the improvements in installation instructions and the code, I've been able to successfully train and apply Cell-TRACTR on the "moma" dataset, after fixing two (minor) code problems:

First, during training the following error occurred:

```

File ".../Cell-TRACTR/src/trackformer/datasets/mot.py", line 101, in __getitem__

man_track_id = self.coco.imgs[idx]['man_track_id']

KeyError: 'man_track_id'

```

I could fix this by adding the following line in the script which transforms datasets from CTC to COCO format, scripts/create_coco_dataset_from_CTC.py, after line 98 (relative to commit 32d37554b2cb9b83b271d5ea0f163f6b29250df6):

`'man_track_id': f'{dataset_name}',

Second, also during training there was a problem with the `datapath` in src/pipeline.py, which I could fix by changing line 127 from

`datapath = args.data_dir / 'CTC_datasets' / dataset / 'CTC' / 'test'`

to

`datapath = args.data_dir / dataset / 'CTC' / 'test'`

Given that all of my points have been addressed by authors and the code problems above are minor and easily fixed, I recommend the manuscript for publication.

Reviewer #3: While the authors have addressed some concerns raised during the previous review, key issues remain unresolved, and the additional experiments fail to strengthen the manuscript’s contributions.

**Major comments:**

- Computational Inefficiency and Resource Intensity: Despite the added discussion on computational constraints, the reliance on an A100 GPU for training for images with the size of 584x600, which is not that large in the broad scope of microscopy. Thus, the method is impractical for widespread adoption unless they show their pre-trained weights can be generalized to other datasets.

Performance on DeepCell: The results on the DeepCell dataset do not favor the claims that this method's performance is comparable with current approaches such as Trackastra and Caliban. While it could be argued that it works on the Mother Machine dataset, it’s an incredibly limited scenario of 1D tracking because the cells' movement is mostly vertical because of the Mother Machine device.

- Unaddressed Reviewer Comments: Reviewer 2's suggestions for experiments corroborating the authors’ original claims were mainly addressed by tuning down their contributions. While this makes the paper more factual, it highlights how the method's performance could be improved.

In summary, the authors have made efforts to address the reviewers' feedback. However, the

limitations of the method persist, and the application of this method beyond the experiments shown here is questionable due to the availability of more computationally efficient and accurate methods.

**Have the authors made all data and (if applicable) computational code underlying the findings in their manuscript fully available?**

Reviewer #1: Yes

Reviewer #2: Yes

Reviewer #3: Yes

PLOS authors have the option to publish the peer review history of their article (what does this mean? ). If published, this will include your full peer review and any attached files.

**Do you want your identity to be public for this peer review?** For information about this choice, including consent withdrawal, please see our Privacy Policy .

Reviewer #1: **Yes: ** Hyoungjun Park

Reviewer #2: No

Reviewer #3: No

**Figure resubmission:**
---

## [Editor Report · Decision Letter 2]

21 Apr 2025

Dear Dr Dunlop,

We are pleased to inform you that your manuscript 'Cell-TRACTR: A transformer-based model for end-to-end segmentation and tracking of cells' has been provisionally accepted for publication in PLOS Computational Biology.

Best regards,

Feilim Mac Gabhann, Ph.D.

Editor-in-Chief

PLOS Computational Biology

---

## [Editor Report · Acceptance letter]

PCOMPBIOL-D-24-01171R2

Cell-TRACTR: A transformer-based model for end-to-end segmentation and tracking of cells

Dear Dr Dunlop,

I am pleased to inform you that your manuscript has been formally accepted for publication in PLOS Computational Biology. Your manuscript is now with our production department and you will be notified of the publication date in due course.

With kind regards,

Anita Estes
